# THE SHAPE OF DATA:
# INTRINSIC DISTANCE FOR DATA DISTRIBUTIONS

**Anton Tsitsulin**[*†]  **Marina Munkhoeva**[*‡]  **Davide Mottin**[§]  **Panagiotis Karras**[§]

**Alex Bronstein**[¶]  **Ivan Oseledets**[‡]  **Emmanuel Müller**[†]

## ABSTRACT

The ability to represent and compare machine learning models is crucial in order to quantify subtle model changes, evaluate generative models, and gather insights on neural network architectures. Existing techniques for comparing data distributions focus on global data properties such as mean and covariance; in that sense, they are *extrinsic* and *uni-scale*. We develop a first-of-its-kind *intrinsic* and *multi-scale* method for characterizing and comparing data manifolds, using a lower-bound of the spectral Gromov-Wasserstein inter-manifold distance, which compares *all* data moments. In a thorough experimental study, we demonstrate that our method effectively discerns the structure of data manifolds even on unaligned data of different dimensionality, and showcase its efficacy in evaluating the quality of generative models.

## 1 INTRODUCTION

The geometric properties of neural networks provide insights about their internals (Morcos et al., 2018; Wang et al., 2018) and help researchers in the design of more robust models (Arjovsky et al., 2017; Bińkowski et al., 2018). Generative models are a natural example of the need for geometric comparison of distributions. As generative models aim to reproduce the true data distribution $\mathbb{P}_d$ by means of the model distribution $\mathbb{P}_g(\mathbf{z}; \Theta)$, more delicate evaluation procedures are needed. Oftentimes, we wish to compare data lying in entirely different spaces, for example to track model evolution or compare models having different representation space.

In order to evaluate the performance of generative models, past research has proposed several *extrinsic* evaluation measures, most notably the Fréchet (Heusel et al., 2017) and Kernel (Bińkowski et al., 2018) Inception Distances (FID and KID). Such measures only reflect the first two or three moments of distributions, meaning they can be insensitive to global structural problems. We showcase this inadvertence in Figure 1: here FID and KID are insensitive to the global structure of the data distribution. Besides, as FID and KID are based only on *extrinsic* properties they are unable to compare *unaligned* data manifolds.

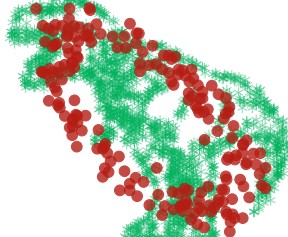

Figure 1: Two distributions having the same first 3 moments, meaning FID and KID scores are close to 0.

In this paper, we start out from the observation that models capturing the *multi-scale* nature of the data manifold by utilizing higher distribution moment matching, such as MMD-GAN (Li et al., 2017) and Sphere-GAN (Park & Kwon, 2019), perform consistently better than their single-scale counterparts. On the other hand, using *extrinsic* information can be misleading, as it is dependent on factors external to the data, such as representation. To address this drawback, we propose IMD, an Intrinsic Multi-scale Distance, that is able to compare distributions using only *intrinsic* information about the data, and provide an efficient approximation thereof that renders computational complexity nearly linear. We demonstrate that IMD effectively quantifies difference in data distributions in three distinct application scenarios: comparing word vectors in languages with *unaligned* vocabularies, tracking dynamics of intermediate neural network representations, and evaluating generative models.

---
[*]Equal contribution. Contact `tsitsulin@bit.uni-bonn.de`, `marina.munkhoeva@skoltech.ru`
[†]University of Bonn   [‡]Skoltech   [§]Aarhus University   [¶]Technion

## 2 RELATED WORK

The geometric perspective on data is ubiquitous in machine learning. Geometric techniques enhance unsupervised and semi-supervised learning, generative and discriminative models (Belkin & Niyogi, 2002; Arjovsky et al., 2017; Mémoli, 2011). We outline the applications of the proposed manifold comparison technique and highlight the geometric intuition along the way.

### 2.1 GENERATIVE MODEL EVALUATION

Past research has explored many different directions for the evaluation of generative models. Setting aside models that ignore the true data distribution, such as the Inception Score (Salimans et al., 2016) and GILBO (Alemi & Fischer, 2018), we discuss most relevant geometric ideas below; we refer the reader to Borji (2019) for a comprehensive survey.

**Critic model-based metrics.** Classifier two-sample tests (C2ST) (Lopez-Paz & Oquab, 2017) aim to assess whether two samples came from the same distribution by means of an auxiliary classifier. This idea is reminiscent of the GAN discriminator network (Goodfellow et al., 2014): if it is possible to train a model that distinguishes between samples from the model and the data distributions, it follows that these distributions are not entirely similar. The convergence process of the GAN-like discriminator (Arjovsky et al., 2017; Bińkowski et al., 2018) lends itself to creating a family of metrics based on training a discriminative classifier (Im et al., 2018). Still, training a separate critic model is often computationally prohibitive and requires careful specification. Besides, if the critic model is a neural network, the resulting metric lacks interpretability and training stability.

Many advanced GAN models such as Wasserstein, MMD, Sobolev and Spherical GANs impose different constraints on the function class so as to stabilize training (Arjovsky et al., 2017; Bińkowski et al., 2018; Mroueh et al., 2018; Park & Kwon, 2019). Higher-order moment matching (Bińkowski et al., 2018; Park & Kwon, 2019) enhances GAN performance, enabling GANs to capture multi-scale data properties, while multi-scale noise ameliorates GAN convergence problems (Jenni & Favaro, 2019). Still, no feasible multi-scale GAN evaluation metric has been proposed to date.

**Positional distribution comparison.** In certain settings, it is acceptable to assign zero probability mass to the real data points (Odena et al., 2018). In effect, metrics that estimate a distribution's location and dispersion provide useful input for generative model evaluations. For instance, the Fréchet Inception Distance (FID) (Heusel et al., 2017) computes the Wasserstein-2 (i.e., Fréchet) distance between distributions approximated with Gaussians, using only the estimated mean and covariance matrices; the Kernel Inception Distance (KID) (Bińkowski et al., 2018) computes a polynomial kernel $\mathrm{k}(x, y) = (\frac{1}{d}x^\top y + 1)^3$ and measures the associated Kernel Maximum Mean Discrepancy (kernel MMD). Unlike FID, KID has an unbiased estimator (Gretton et al., 2012; Bińkowski et al., 2018). However, even while such methods, based on a limited number of moments, may be computationally inexpensive, they only provide a rudimentary characterization of distributions from a geometric viewpoint.

**Intrinsic geometric measures.** The Geometry Score (Khrulkov & Oseledets, 2018) characterizes distributions in terms of their estimated persistent homology, which roughly corresponds to the number of holes in a manifold. Still, the Geometry Score assesses distributions merely in terms of their *global* geometry. In this work, we aim to provide a *multi-scale* geometric assessment.

### 2.2 SIMILARITIES OF NEURAL NETWORK REPRESENTATIONS

Learning how representations evolve during training or across initializations provides a pathway to the interpretability of neural networks (Raghu et al., 2017). Still, state-of-the-art methods for comparing representations of neural networks (Kornblith et al., 2019; Morcos et al., 2018; Wang et al., 2018) consider only linear projections. The intrinsic nature of IMD renders it appropriate for the task of comparing neural network representations, which can only rely on intrinsic information.

Yin & Shen (2018) introduced the Pairwise Inner Product (PIP) loss, an unnormalized covariance error between sets, as a dissimilarity metric between word2vec embedding spaces with common vocabulary. We show in Section 4.2 how IMD is applicable to this comparison task too.

## 3 MULTI-SCALE INTRINSIC DISTANCE

At the core of deep learning lies the *manifold hypothesis*, which states that high-dimensional data, such as images or text, lie on a low-dimensional manifold (Narayanan & Mitter, 2010; Belkin & Niyogi, 2002; 2007). We aim to provide a theoretically motivated comparison of data manifolds based on rich intrinsic information. Our target measure should have the following properties:

**intrinsic** – it is invariant to isometric transformations of the manifold, e.g. translations or rotations.

**multi-scale** – it captures both local and global information.

We expose our method starting out with heat kernels, which admit a notion of manifold metric and can be used to lower-bound the distance between manifolds.

### 3.1 HEAT KERNELS ON MANIFOLDS AND GRAPHS

Based on the heat equation, the heat kernel captures *all* the information about a manifold's intrinsic geometry (Sun et al., 2009). Given the Laplace-Beltrami operator (LBO) $\Delta_{\mathcal{X}}$ on a manifold $\mathcal{X}$, the *heat equation* is $\frac{\partial u}{\partial t} = \Delta_{\mathcal{X}} u$ for $u : \mathbb{R}^+ \times \mathcal{X} \to \mathbb{R}^+$. A smooth function $u$ is a *fundamental solution* of the heat equation at point $x \in \mathcal{X}$ if $u$ satisfies both the heat equation and the Dirac condition $u(t, x') \to \delta(x' - x)$ as $t \to 0^+$. We assume the Dirichlet boundary condition $u(t, x) = 0$ for all $t$ and $x \in \partial \mathcal{X}$. The heat kernel $\mathrm{k}_{\mathcal{X}} : \mathcal{X} \times \mathcal{X} \times \mathbb{R}^+ \to \mathbb{R}_0^+$ is the unique solution of the heat equation; while heat kernels can be defined on hyperbolic spaces and other exotic geometries, we restrict our exposition to Euclidean spaces $\mathcal{X} = \mathbb{R}^d$, on which the heat kernel is defined as:

$$\mathrm{k}_{\mathbb{R}^d}(x, x', t) = \frac{1}{(4\pi t)^{d/2}} \exp\left(-\frac{\|x - x'\|^2}{4t}\right) \tag{1}$$

For a compact $\mathcal{X}$ including submanifolds of $\mathbb{R}^d$, the heat kernel admits the expansion $\mathrm{k}_{\mathcal{X}}(x, x', t) = \sum_{i=0}^{\infty} e^{-\lambda_i t} \phi_i(x) \phi_i(x')$, where $\lambda_i$ and $\phi_i$ are the $i$-th eigenvalue and eigenvector of $\Delta_{\mathcal{X}}$. For $t \simeq 0^+$, according to Varadhan's lemma, the heat kernel approximates geodesic distances. Importantly for our purposes, the Heat kernel is *multi-scale*: for a local domain $\mathcal{D}$ with Dirichlet condition, the localized heat kernel $\mathrm{k}_{\mathcal{D}}(x, x', t)$ is a good approximation of $\mathrm{k}_{\mathcal{X}}(x, x', t)$ if either (i) $\mathcal{D}$ is arbitrarily small and $t$ is small enough, or (ii) $t$ is for arbitrarily large and $\mathcal{D}$ is big enough. Formally,

**Definition 1 Multi-scale property** *(Grigor'yan, 2006; Sun et al., 2009) (i) For any smooth and relatively compact domain $\mathcal{D} \subseteq \mathcal{X}$, $\lim_{t \to 0} \mathrm{k}_{\mathcal{D}}(x, x', t) = \mathrm{k}_{\mathcal{X}}(x, x', t)$ (ii) For any $t \in \mathbb{R}^+$ and any $x, x' \in \mathcal{D}_1$ localized heat kernel $\mathrm{k}_{\mathcal{D}_1}(x, x', t) \leq \mathrm{k}_{\mathcal{D}_2}(x, x', t)$ if $\mathcal{D}_1 \subseteq \mathcal{D}_2$. Moreover, if $\{\mathcal{D}_n\}$ is an expanding and exhausting sequence $\bigcup_{i=1}^{\infty} \mathcal{D}_i = \mathcal{X}$ and $\mathcal{D}_{i-1} \subseteq \mathcal{D}_i$, then $\lim_{i \to \infty} \mathrm{k}_{\mathcal{D}_i}(x, x', t) = \mathrm{k}_{\mathcal{X}}(x, x', t)$ for any $t$.*

Heat kernels are also defined for graphs in terms of their Laplacian matrices. An undirected graph is a pair $G = (V, E)$, where $V = (v_1, \ldots, v_n), n = |V|$, is the set of vertices and $E \subseteq (V \times V)$ the set of edges. The *adjacency matrix* of $G$ is a $n \times n$ matrix $\mathbf{A}$ having $\mathbf{A}_{ij} = 1$ if $(i, j) \in E$ and $A_{ij} = 0$ otherwise. The *normalized graph Laplacian* is the matrix $\mathcal{L} = \mathbf{I} - \mathbf{D}^{-\frac{1}{2}} \mathbf{A} \mathbf{D}^{-\frac{1}{2}}$, where $\mathbf{D}$ is the diagonal matrix in which entry $\mathbf{D}_{ii}$ holds the degree of node $i$, i.e, $\mathbf{D}_{ii} = \sum_{j=1}^{n} \mathbf{A}_{ij}$. Since the Laplacian matrix is symmetric, its eigenvectors $\phi_1, \ldots, \phi_n$, are real and orthonormal. Thus, it is factorized as $\mathcal{L} = \mathbf{\Phi} \mathbf{\Lambda} \mathbf{\Phi}^{\top}$, where $\mathbf{\Lambda}$ is a diagonal matrix with the sorted eigenvalues $\lambda_1 \leq \ldots \leq \lambda_n$, and $\Phi$ is the orthonormal matrix $\mathbf{\Phi} = (\phi_1, \ldots, \phi_n)$ having the eigenvectors of $\mathcal{L}$ as its columns. The heat kernel on a graph is also given by the solution to the heat equation on a graph, which requires an eigendecomposition of its Laplacian: $\mathbf{H}_t = e^{-t\mathcal{L}} = \mathbf{\Phi} e^{-t\mathbf{\Lambda}} \mathbf{\Phi}^{\top} = \sum_i e^{-t\lambda_i} \phi_i \phi_i^{\top}$.

A useful invariant of the heat kernel is the *heat kernel trace* $\mathrm{hkt}_{\mathcal{X}} : \mathcal{X} \times \mathbb{R}_0^+ \to \mathbb{R}_0^+$, defined by a diagonal restriction as $\mathrm{hkt}_{\mathcal{X}}(t) = \int_{\mathcal{X}} \mathrm{k}_{\mathcal{X}}(x, x, t) dx = \sum_{i=0}^{\infty} e^{-\lambda_i t}$ or, in the discrete case, $\mathrm{hkt}_{\mathcal{L}}(t) = \mathrm{Tr}(\mathbf{H}_t) = \sum_i e^{-t\lambda_i}$. Heat kernels traces (HKTs) have been successfully applied to the analysis of 3D shapes (Sun et al., 2009) and graphs (Tsitsulin et al., 2018). The HKT contains *all* the information in the graph's spectrum, both local and global, as the eigenvalues $\lambda_i$ can be inferred therefrom (Mémoli, 2011, Remark 4.8). For example, if there are $c$ connected components in the graph, then $\lim_{t \to \infty} \mathrm{hkt}_{\mathcal{L}}(t) = c$.

## 3.2 Convergence to the Laplace-Beltrami Operator

An important property of graph Laplacians is that it is possible to construct a graph among points sampled from a manifold $\mathcal{X}$ such that the spectral properties of its Laplacian resemble those of the Laplace-Beltrami operator on $\mathcal{X}$. Belkin and Niyogi (Belkin & Niyogi, 2002) proposed such a construction, the point cloud Laplacian, which is used for dimensionality reduction in a technique called Laplacian eigenmaps. Convergence to the LBO has been proven for various definitions of the graph Laplacian, including the one we use (Belkin & Niyogi, 2007; Hein et al., 2007; Coifman & Lafon, 2006; Ting et al., 2010). We recite the convergence results for the point cloud Laplacian from Belkin & Niyogi (2007):

**Theorem 1** *Let $\lambda_{n,i}^{t_n}$ and $\phi_{n,i}^{t_n}$ be the $i^{\text{th}}$ eigenvalue and eigenvector, respectively, of the point cloud Laplacian $\mathcal{L}^{t_n}$; let $\lambda_i$ and $\phi_i$ be the $i^{\text{th}}$ eigenvalue and eigenvector of the LBO $\Delta$. Then, there exists $t_n \to 0$ such that*

$$\lim_{n \to \infty} \lambda_{n,i}^{t_n} = \lambda_i$$
$$\lim_{n \to \infty} \left\| \phi_{n,i}^{t_n} - \phi_i \right\|_2 = 0$$

Still, the point cloud Laplacian involves the creation of an $\mathcal{O}(n^2)$ matrix; for the sake of scalability, we use the $k$-nearest-neighbours ($k$NN) graph by OR-construction (i.e., based on bidirectional $k$NN relationships among points), whose Laplacian converges to the LBO for data with sufficiently high intrinsic dimension (Ting et al., 2010). As for the choice of $k$, a random geometric $k$NN graph is connected when $k \geq \log n/\log 7 \approx 0.5139 \log n$ (Balister et al., 2005); $k = 5$ yields connected graphs for all sample sizes we tested.

## 3.3 Spectral Gromov-Wasserstein Distance

Even while it is a multi-scale metric *on* manifolds, the heat kernel can be spectrally approximated by finite graphs constructed from points sampled from these manifolds. In order to construct a metric *between* manifolds, Mémoli (2011) suggests an optimal-transport-theory-based "meta-distance": a spectral definition of the Gromov-Wasserstein distance between Riemannian manifolds based on matching the heat kernels at all scales. The cost of matching a pair of points $(x, x')$ on manifold $\mathcal{M}$ to a pair of points $(y, y')$ on manifold $\mathcal{N}$ at scale $t$ is given by their heat kernels $k_\mathcal{M}, k_\mathcal{N}$:

$$\Gamma(x, y, x', y', t) = \left| k_\mathcal{M}(x, x', t) - k_\mathcal{N}(y, y', t) \right|.$$

The distance between the manifolds is then defined in terms of the infimal measure coupling

$$d_{\text{GW}}(\mathcal{M}, \mathcal{N}) = \inf_\mu \sup_{t>0} e^{-2(t+t^{-1})} \left\| \Gamma \right\|_{L^2(\mu \times \mu)},$$

where the infimum is sought over all measures $\mu$ on $\mathcal{M} \times \mathcal{N}$ marginalizing to the standard measures on $\mathcal{M}$ and $\mathcal{N}$. For finite spaces, $\mu$ is a doubly-stochastic matrix. This distance is lower-bounded (Mémoli, 2011) in terms of the respective heat kernel traces as:

$$d_{\text{GW}}(\mathcal{M}, \mathcal{N}) \geq \sup_{t>0} e^{-2(t+t^{-1})} \left| \text{hkt}_\mathcal{M}(t) - \text{hkt}_\mathcal{N}(t) \right|. \tag{2}$$

This lower bound is the scaled $L_\infty$ distance between the *heat trace signatures* $\text{hkt}_\mathcal{M}$ and $\text{hkt}_\mathcal{N}$. The scaling factor $e^{-2(t+t^{-1})}$ favors medium-scale differences, meaning that this lower bound is not sensitive to local perturbations. The maximum of the scaling factor occurs at $t = 1$, and more than $1 - 10^{-8}$ of the function mass lies between $t = 0.1$ and $t = 10$.

## 3.4 Heat Trace Estimation

Calculating the heat trace signature efficiently and accurately is a challenge on a large graph as it involves computing a trace of a large matrix exponential, i.e. $\text{Tr}(e^{-t\mathcal{L}})$. A naive approach would be to use an eigendecomposition $\exp(-t\mathcal{L}) = \Phi \exp(-t\Lambda)\Phi^\top$, which is infeasible for large $n$. Recent work (Tsitsulin et al., 2018) suggested using either truncated Taylor expansion or linear interpolation of the interloping eigenvalues, however, both techniques are quite coarse. To combine

accuracy and speed, we use the Stochastic Lanczos Quadrature (SLQ) (Ubaru et al., 2017; Golub & Meurant, 2009). This method combines the Hutchinson trace estimator (Hutchinson, 1989; Adams et al., 2018) and the Lanczos algorithm for eigenvalues. We aim to estimate the trace of a matrix function with a Hutchinson estimator:

$$\mathrm{Tr}(f(\mathcal{L})) = \mathbb{E}_{p(\mathbf{v})}(\mathbf{v}^\top f(\mathcal{L})\mathbf{v}) \approx \frac{n}{n_v} \sum_{i=1}^{n_v} \mathbf{v}_i^\top f(\mathcal{L})\mathbf{v}_i, \tag{3}$$

where the function of interest $f(\cdot) = \exp(\cdot)$ and $\mathbf{v}_i$ are $n_v$ random vectors drawn from a distribution $p(\mathbf{v})$ with zero mean and unit variance. A typical choice for $p(\mathbf{v})$ is Rademacher or a standard normal distribution. In practice, there is little difference, although in theory Rademacher has less variance, but Gaussian requires less random vectors (Avron & Toledo, 2011).

To estimate the quadratic form $\mathbf{v}_i^\top f(\mathcal{L})\mathbf{v}_i$ in (3) with a symmetric real-valued matrix $\mathcal{L}$ and a smooth function $f$, we plug in the eigendecomposition $\mathcal{L} = \Phi\Lambda\Phi^\top$, rewrite the outcome as a Riemann-Stieltjes integral and apply the $m$-point Gauss quadrature rule (Golub & Welsch, 1969):

$$\mathbf{v}_i^\top f(\mathcal{L})\mathbf{v}_i = \mathbf{v}_i^\top \Phi f(\Lambda)\Phi^\top \mathbf{v}_i = \sum_{j=1}^{n} f(\lambda_j)\mu_j^2 = \int_a^b f(t)d\mu(t) \approx \sum_{k=1}^{m} \omega_k f(\theta_k), \tag{4}$$

where $\mu_j = [\Phi^\top \mathbf{v}_i]_j$ and $\mu(t)$ is a piecewise constant function defined as follows

$$\mu(t) = \begin{cases} 0, & \text{if } t < a = \lambda_n \\ \sum_{j=1}^{i} \mu_j^2, & \text{if } \lambda_i \leq t < \lambda_{i-1} \\ \sum_{j=1}^{n} \mu_j^2, & \text{if } b = \lambda_1 \leq t \end{cases}$$

and $\theta_k$ are the quadrature's nodes and $\omega_k$ are the corresponding weights. We obtain $\omega_k$ and $\theta_k$ with the $m$-step Lanczos algorithm (Golub & Meurant, 2009), which we describe succinctly.

Given the symmetric matrix $\mathcal{L}$ and an arbitrary *starting unit-vector* $\mathbf{q}_0$, the $m$-step Lanczos algorithm computes an $n \times m$ matrix $\mathbf{Q} = [\mathbf{q}_0, \mathbf{q}_1, \ldots, \mathbf{q}_{m-1}]$ with orthogonal columns and an $m \times m$ tridiagonal symmetric matrix $\mathbf{T}$, such that $\mathbf{Q}^\top\mathcal{L}\mathbf{Q} = \mathbf{T}$. The columns of $\mathbf{Q}$ constitute an orthonormal basis for the Krylov subspace $\mathcal{K}$ that spans vectors $\{\mathbf{q}_0, \mathcal{L}\mathbf{q}_0, \ldots, \mathcal{L}^{m-1}\mathbf{q}_0\}$; each $\mathbf{q}_i$ vector is given as a polynomial in $\mathcal{L}$ applied to the initial vector $\mathbf{q}_0$: $\mathbf{q}_i = p_i(\mathcal{L})\mathbf{q}_0$. These Lanczos polynomials are orthogonal with respect to the integral measure $\mu(t)$. As orthogonal polynomials satisfy the three term recurrence relation, we obtain $p_{k+1}$ as a combination of $p_k$ and $p_{k-1}$. The tridiagonal matrix storing the coefficients of such combinations, called the Jacobi matrix $\mathbf{J}$, is exactly the tridiagonal symmetric matrix $\mathbf{T}$. A classic result tells us that the nodes $\theta_k$ and the weights $\omega_k$ of the Gauss quadrature are the eigenvalues of $\mathbf{T}$, $\lambda_k$, and the squared first components of its normalized eigenvectors, $\tau_k^2$, respectively (see Golub & Welsch (1969); Wilf (1962); Golub & Meurant (2009)). Thereby, setting $\mathbf{q}_0 = \mathbf{v}_i$, the estimate for the quadratic form becomes:

$$\mathbf{v}_i^\top f(\mathcal{L})\mathbf{v}_i \approx \sum_{k=1}^{m} \tau_k^2 f(\lambda_k), \quad \tau_k = \mathbf{U}_{0,k} = \mathbf{e}_1^\top \mathbf{u}_k, \quad \lambda_k = \Lambda_{k,k} \quad \mathbf{T} = \mathbf{U}\Lambda\mathbf{U}^\top, \tag{5}$$

Applying (5) over $n_v$ random vectors in the Hutchinson trace estimator (3) yields the SLQ estimate:

$$\mathrm{Tr}(f(\mathcal{L})) \approx \frac{n}{n_v} \sum_{i=1}^{n_v} \left( \sum_{k=0}^{m} \left(\tau_k^i\right)^2 f\left(\lambda_k^i\right) \right) = \Gamma. \tag{6}$$

We derive error bounds for the estimator based on the Lanczos approximation of the matrix exponential, and show that even a few Lanczos steps, i.e., $m = 10$, are sufficient for an accurate approximation of the quadratic form. However, the trace estimation error is theoretically dominated by the error of the Hutchinson estimator, e.g. for Gaussian $p(\mathbf{v})$ the bound on the number of samples to guarantee that the probability of the relative error exceeding $\epsilon$ is at most $\delta$ is $8\epsilon^{-2}\ln(2/\delta)$ (Roosta-Khorasani & Ascher, 2015). Although, in practice, we observe performance much better than the bound suggests. Hutchinson error implies nearing accuracy roughly $10^{-2}$ with $n_v \geq 10k$ random vectors, however, with as much as $n_v = 100$ the error is already $10^{-3}$. Thus, we use default values of $m = 10$ and $n_v = 100$ in all experiments in Section 4. Please see Appendix A for full derivations and figures.

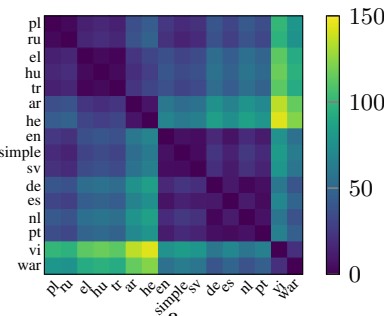 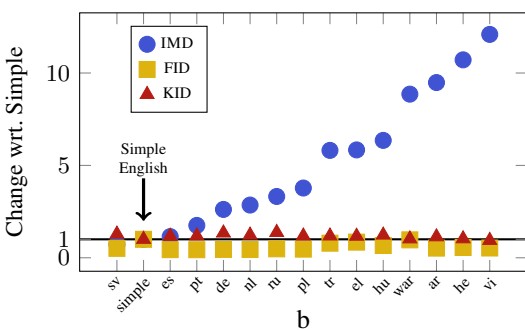

a                                          b

Figure 2: (a) IMD distances between language pairs for unaligned Wikipedia word embeddings and (b) distances from the simple English Wikipedia visualized for IMD, FID, and KID. We consider 16 languages: Polish, Russian, Greek, Hungarian, Turkish, Arabic, Hebrew, English, Simple English, Swedish, German, Spanish, Dutch, Portugese, Vietnamese, and Waray-Waray.

### 3.5 PUTTING IMD TOGETHER

We employ the heretofore described advances in differential geometry and numerical linear algebra to create IMD (*Multi-Scale Intrinsic Distance*), a fast, intrinsic method to lower-bound the spectral Gromov-Wasserstein distance between manifolds.

We describe the overall computation of IMD in Algorithm 1. Given data samples in $\mathbb{R}^d$, we build a $k$NN graph $G$ by OR-construction such that its Laplacian spectrum approximates the one of the Laplace-Beltrami operator of the underlying manifold (Ting et al., 2010), and then compute $\mathrm{hkt}_G(t) = \sum_i e^{-\lambda_i t} \approx \Gamma$. We compare heat traces in the spirit of Equation (2), i.e., $|\mathrm{hkt}_{G_1}(t) - \mathrm{hkt}_{G_2}(t)|$ for $t \in (0.1, 10)$ sampled from a logarithmically spaced grid.

---

**Algorithm 1** IMD algorithm.

**function** IMDESC($X$)
    $G \leftarrow \mathtt{kNN}(X)$
    $\mathcal{L} \leftarrow \mathtt{Laplacian}(G)$
    **return** $\Gamma = \mathtt{slq}(\mathcal{L}, s, n_v)$

**function** IMDIST($X, Y$)
    $\mathrm{hkt}_X \leftarrow \mathtt{IMDist}(X)$
    $\mathrm{hkt}_Y \leftarrow \mathtt{IMDist}(Y)$
    **return** $\sup e^{-2(t+t^{-1})}|\mathrm{hkt}_X - \mathrm{hkt}_Y|$

---

Constructing exact $k$NN graphs is an $\mathcal{O}(dn^2)$ operation; however, approximation algorithms take near-linear time $\mathcal{O}(dn^{1+\omega})$ (Dong et al., 2011; Aumüller et al., 2019). In practice, with approximate $k$NN graph construction (Dong et al., 2011), computational time is low while result variance is similar to the exact case. The $m$-step Lanczos algorithm on a sparse $n \times n$ $k$NN Laplacian $\mathcal{L}$ with one starting vector has $\mathcal{O}(knm)$ complexity, where $kn$ is the number of nonzero elements in $\mathcal{L}$. The symmetric tridiagonal matrix eigendecomposition incurs an additional $\mathcal{O}(m \log m)$ (Coakley & Rokhlin, 2013). We apply this algorithm over $n_v$ starting vectors, yielding a complexity of $\mathcal{O}(n_v(m \log m + kmn))$, with constant $k = 5$ and $m = 10$ by default. In effect, IMD's time complexity stands between those of two common GAN evaluation methods: KID, which is $\mathcal{O}(dn^2)$ and FID, which is $\mathcal{O}(d^3 + dn)$. The time complexity of Geometry Score is unspecified in Khrulkov & Oseledets (2018), yet in Section 4.6 we show that its runtime grows exponentially in sample size.

## 4 EXPERIMENTS

We evaluate IMD on the ability to compare intermediate representations of machine learning models. For instance, in a recommender system we could detect whether a problem is related to the representation or the the classifier in the end of a pipeline. In this section, we show the effectiveness of our intrinsic measure on multiple tasks and show how our intrinsic distance can provide insights beyond previously proposed extrinsic measures.

**Summary of experiments.** We examine the ability of IMD[1] to measure several aspects of difference among data manifolds. We first consider a task from unsupervised machine translation with unaligned word embeddings and show that IMD captures correlations among language kinship (affinity or genealogical relationships). Second, we showcase how IMD handles data coming from data sources of unequal dimensionalities. Third, we study how IMD highlights differences among image data representations across initializations and through training process of neural networks.

---

[1] Our code is available open-source: https://github.com/xgfs/imd.

## 4.1 COMPARING UNALIGNED LANGUAGE MANIFOLDS

The problem of unaligned representations is particularly severe in the domain of natural language processing as the vocabulary is rarely comparable across different languages or even different documents. We employ IMD to measure the relative closeness of pairs of languages based on the word embeddings with different vocabularies. Figure 2 (a) shows a heatmap of pairwise IMD scores. IMD detects similar languages (Slavic, Semitic, Romanic, etc.) despite the lack of ground truth vocabulary alignment. On the other hand, Figure 11 in Appendix C shows that FID and KID, are not able to distinguish the intrinsic language-specific structure in word embeddings. Detailed description and setting of the experiment can be found in Appendix C.

## 4.2 OPTIMIZING DIMENSIONALITY OF WORD EMBEDDINGS

Comparing data having different dimensionality is cumbersome, even when representations *are* aligned. We juxtapose IMD by PIP loss (Yin & Shen, 2018) which allows the comparison of aligned representations for word embeddings. To this end, we measure IMD distance between English word embeddings of varying dimensions. Figure 3 shows the heatmap of the scores between sets of word vectors of different dimensionalities. Closer dimensionalities have lower distance scores for both metrics. However, IMD better highlights gradual change of the size of word

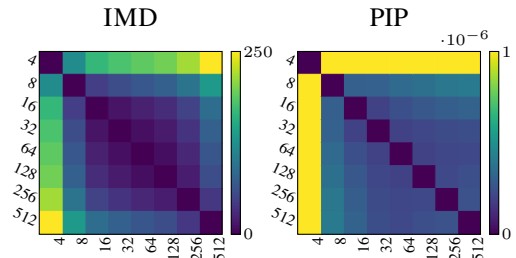

Figure 3: Comparison of IMD and PIP loss on word embeddings of different dimension. IMD detects subtle changes in the dimensionality.

vectors, e.g. word vectors of size 4 and 8 are clearly closer to each other than embeddings of size 4 and 16 in terms of IMD, which is not true for PIP.

## 4.3 TRACKING THE EVOLUTION OF IMAGE MANIFOLDS

Next, we employ IMD to inspect the internal dynamics of neural networks. We investigate the stability of output layer manifolds across random initializations. We train 10 instances of the VGG-16 (Simonyan & Zisserman, 2015) network using different weight initializations on the CIFAR-10 and CIFAR-100 datasets. We compare the average IMD scores across representations in each network layer relative to the last layer. As Figure 4 (left) shows, for both CIFAR-10 and CIFAR-100, the convolutional layers exhibit similar behavior; IMD shows that consequent layers do not monotonically contribute to the separation of image representations, but start to do so after initial feature extraction stage comprised of 4 convolutional blocks. A low variance across the 10 networks trained from different random initializations indicates stability in the network structure.

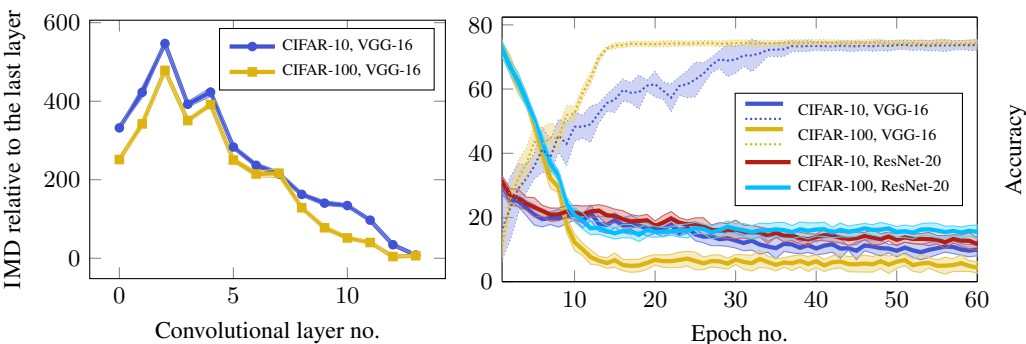

Figure 4: *(left)* IMD score across convolutional layers of the VGG-16 network on CIFAR-10 and CIFAR-100 datasets; *(right)* training progression in terms of accuracy (dotted) and IMD (solid) on CIFAR-10 and CIFAR-100 datasets for VGG-16 and ResNet-20, with respect to VGG-16.

Table 1: IMD agrees with KID and FID across varying datasets for GAN evaluation.

| Metric | MNIST | | FashionMNIST | | CIFAR10 | | CelebA | |
|---|---|---|---|---|---|---|---|---|
| | WGAN | WGAN-GP | WGAN | WGAN-GP | WGAN | WGAN-GP | WGAN | WGAN-GP |
| IMD | $57.74 \pm$ 0.47 | $10.77 \pm$ 0.42 | $118.14 \pm$ 0.52 | $13.45 \pm$ 0.54 | $18.10 \pm$ 0.36 | $10.84 \pm$ 0.42 | $10.11 \pm$ 0.33 | $2.84 \pm$ 0.31 |
| KID $\times 10^3$ | $47.26 \pm$ 0.07 | $5.53 \pm$ 0.03 | $119.93 \pm$ 0.14 | $25.49 \pm$ 0.07 | $93.89 \pm$ 0.09 | $59.59 \pm$ 0.09 | $217.28 \pm$ 0.14 | $92.71 \pm$ 0.08 |
| FID | $31.75 \pm$ 0.07 | $8.95 \pm$ 0.03 | $152.44 \pm$ 0.12 | $35.31 \pm$ 0.07 | $101.43 \pm$ 0.09 | $80.65 \pm$ 0.09 | $205.63 \pm$ 0.09 | $85.55 \pm$ 0.08 |

We now examine the last network layers during training with different initializations. Figure 4 (right) plots the VGG16 validation errors and IMD scores relative to the final layer representations of *two* pretrained networks, VGG16 itself with last layer dimension $d = 512$ and ResNet-20 with $d = 64$ and $\sim 50$ times less parameters. We observe that even in such unaligned spaces, IMD correctly identifies the convergence point of the networks. Surprisingly, we find that, in terms of IMD, VGG-16 representations progress towards not only the VGG-16 final layer, but the ResNet-20 final layer representation as well; this result suggests that these networks of distinct architectures share similar final structures.

## 4.4 Evaluating Generative Models

We now move on to apply IMD to evaluation of generative models. First, we evaluate the sensitivity of IMD, FID, and KID to simple image transformations as a proxy to more intricate artifacts of modern generative models. We progressively blur images from the CIFAR-10 training set, and measure the distance to the original data manifold, averaging outcomes over 100 subsamples of 10k images each. To enable comparison across methods, we normalize each distance measure such that the distance between CIFAR-10 and MNIST is 1. Figure 5 reports the results at different levels $\sigma$ of Gaussian blur. We additionally report the normalized distance to the CIFAR-100 training set (dashed lines ). FID and KID quickly drift away from the original distribution and match MNIST, a dataset of a completely different

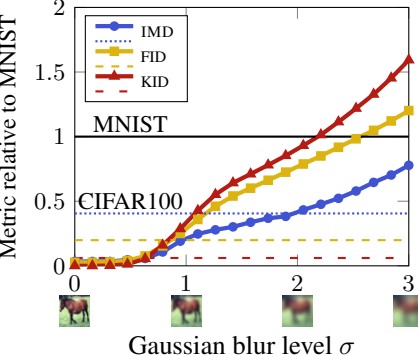

Figure 5: FID, KID and IMD on the CIFAR-10 dataset with Gaussian blur.

nature. Contrariwise, IMD is more robust to noise and follows the datasets structure, as the relationships between objects remain mostly unaffected on low blur levels. Moreover, with both FID and KID, low noise ($\sigma = 1$) applied to CIFAR-10 suffices to exceed the distance of CIFAR-100, which is similar to CIFAR-10. IMD is much more robust, exceeding that distance only with $\sigma = 2$.

Next, we turn our attention to the sample-based evaluation of generative models. We then train the WGAN (Arjovsky et al., 2017) and WGAN-GP (Gulrajani et al., 2017) models on four datasets: MNIST, FashionMNIST, CIFAR10 and CelebA. We sample 10k samples, **Y**, from each GAN. We then uniformly subsample $10k$ images from the corresponding original dataset, **X**, and compute the IMD, KID and FID scores between **X** and **Y**. Table 1 reports the average measure and its 99% confidence interval across 100 runs. IMD, as well as both FID and KID, reflect the fact that WGAN-GP is a more expressive model. We provide details on architecture, training, and generated samples in Appendix C. Additionally, in Appendix C we demonstrate superiority of IMD on synthetic data.

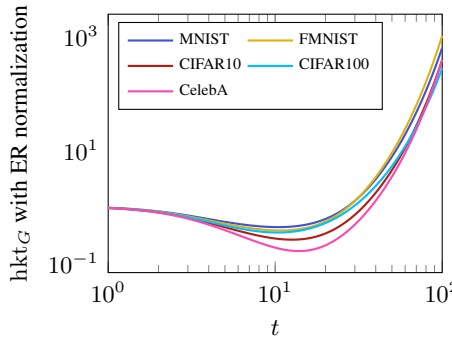

Figure 6: Plotting the normalized heat trace allows interpretation of medium- and global-scale structure of datasets. Best viewed in color.

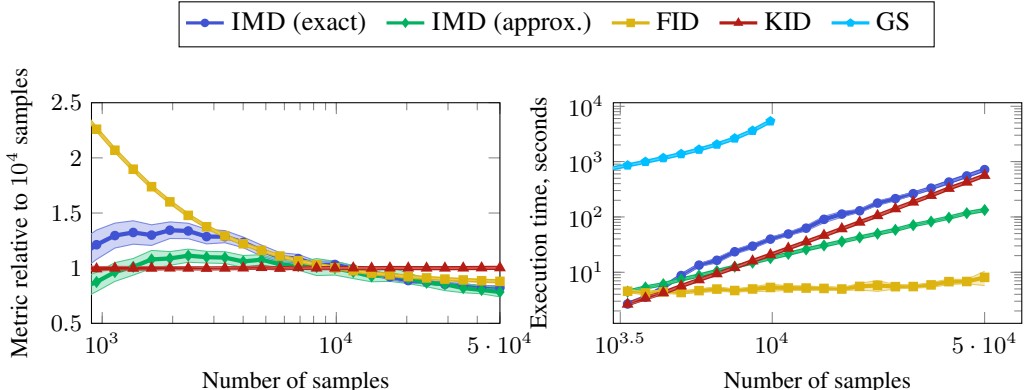

Figure 7: Stability and scalability experiment: *(left)* stability of FID, KID and IMD wrt. sample size on CIFAR-10 and CIFAR-100 dataset; *(right)* scalability of FID, KID and IMD wrt. sample size on synthetic datasets.

### 4.5 INTERPRETING IMD

To understand how IMD operates, we investigate the behavior of heat kernel traces of different datasets that are normalized by a null model. Tsitsulin et al. (2018) proposed a normalization by the heat kernel trace of an empty graph, which amounts to taking the average, rather than the sum, of the original heat kernel diagonal. However, this normalization is not an appropriate null model as it ignores graph connectivity. We propose a heat kernel normalization by the *expected* heat kernel of an Erdős-Rényi graph (further details in the Appendix D).

Figure 6 depicts the obtained normalized $\mathrm{hkt}_g$ for all datasets we work with. We average results over 100 subsamples of $10k$ images each. For $t = 10$, i.e., at a medium scale, CelebA is most different from the random graph, while for large-scale $t$ values, which capture global community structure, $\frac{\mathrm{dhkt}_g(t)}{\mathrm{d}t}$ reflects the approximate number of clusters in the data. Surprisingly, CIFAR-100 comes close to CIFAR-10 for large $t$ values; we have found that this is due to the fact that the pre-trained Inception network does not separate the CIFAR-100 data classes well enough. We conclude that the heat kernel trace is interpretable if we normalize it with an appropriate null model.

### 4.6 VERIFYING STABILITY AND SCALABILITY OF IMD

In addition to the complexity analysis in Section 3.5, we assess the scaling and sample stability of IMD. Since IMD, like FID, is a lower bound to an optimal transport-based metric, we cannot hope for an unbiased estimator. However, we empirically verify, in Figure 7 (left), that IMD does not diverge too much with increased sample size. Most remarkably, we observe that IMD with approximate $k$NN (Dong et al., 2011) does not induce additional variance, while it diverges slightly further than the exact version as the number of samples grows.

In terms of scalability, Figure 7 (right) shows that the theoretical complexity is supported in practice. Using approximate $k$NN, we break the $\mathcal{O}(n^2)$ performance of KID. FID's time complexity appears constant, as its runtime is dominated by the $\mathcal{O}(d^3)$ matrix square root operation. Geometry score (GS) fails to perform scalably, as its runtime grows exponentially. Due to this prohibitive computational cost, we eschew other comparison with GS. Furthermore, as IMD distance is computed through a low-dimensional heat trace representation of the manifold, we can store HKT for future comparisons, thereby enhancing performance in the case of many-to-many comparisons.

## 5 DISCUSSION AND FUTURE WORK

We introduced IMD, a geometry-grounded, first-of-its-kind intrinsic multi-scale method for comparing unaligned manifolds, which we approximate efficiently with guarantees, utilizing the Stochastic Lanczos Quadrature. We have shown the expressiveness of IMD in quantifying the change of data representations in NLP and image processing, evaluating generative models, and in the study of neural network representations. Since IMD allows comparing diverse manifolds, its applicability is not limited to the tasks we have evaluated, while it paves the way to the development of even more expressive techniques founded on geometric insights.

## ACKNOWLEDGEMENTS

This work was partially funded by the Ministry of Science and Education of Russian Federation as a part of Mega Grant Research Project 14.756.31.0001. Ivan Oseledets would like to thank Huawei for the support of his research.

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

APPENDIX

## A   TRACE ESTIMATION ERROR BOUNDS

We will use the error of the Lanczos approximation of the action of the matrix exponential $f(\mathcal{L})\mathbf{v} = \exp^{-t\mathcal{L}}\mathbf{v}$ to estimate the error of the trace. We first rewrite quadratic form under summation in the trace approximation to a convenient form,

$$\mathbf{v}^\top f(\mathcal{L})\mathbf{v} \approx \sum_{k=0}^m \tau_k^2 f(\lambda_k) = \sum_{k=0}^m [\mathbf{e}_1^\top \mathbf{u}_k]^2 f(\lambda_k) = \mathbf{e}_1^\top \mathbf{U} f(\Lambda)\mathbf{U}^\top \mathbf{e}_1 = \mathbf{e}_1^\top f(\mathbf{T})\mathbf{e}_1. \quad (7)$$

Because the Krylov subspace $\mathcal{K}_m(\mathcal{L}, \mathbf{v})$ is built on top of vector $\mathbf{v}$ with $\mathbf{Q}$ as an orthogonal basis of $\mathcal{K}_m(\mathcal{L}, \mathbf{v})$, i.e. $\mathbf{q}_0 = \mathbf{v}$ and $\mathbf{v} \perp \mathbf{q}_i$ for $i \in (1, \dots, m-1)$, the following holds

$$\mathbf{v}^\top f(\mathcal{L})\mathbf{v} \approx \mathbf{v}^\top \mathbf{Q} f(\mathbf{T})\mathbf{e}_1 = \mathbf{e}_1^\top f(\mathbf{T})\mathbf{e}_1. \quad (8)$$

Thus, the error in quadratic form estimate $\mathbf{v}^\top f(\mathcal{L})\mathbf{v}$ is exactly the error of Lanczos approximation $f(\mathcal{L})\mathbf{v} \approx \mathbf{Q}f(\mathbf{T})\mathbf{e}_1$. To obtain the error bounds, we use the Theorem 2 in Hochbruck & Lubich (1997), which we recite below.

**Theorem 2** *Let $\mathcal{L}$ be a real symmetric positive semi-definite matrix with eigenvalues in the interval $[0, 4\rho]$. Then the error in the m-step Lanczos approximation of $\exp^{-t\mathcal{L}}\mathbf{v}$, i.e. $\epsilon_m = \|\exp^{-t\mathcal{L}}\mathbf{v} - \mathbf{Q}_m \exp^{-t\mathbf{T}_m}\mathbf{e}_1\|$, is bounded in the following ways:*

$$\epsilon_m \leq \begin{cases} 10 e^{-m^2/(5\rho t)}, & \sqrt{4\rho t} \leq m \leq 2\rho t & (9a) \\ 10(\rho t)^{-1} e^{-\rho t} \left(\dfrac{e\rho t}{m}\right)^m, & m \geq 2\rho t & (9b) \end{cases}$$

Since $\mathbf{v}$ is a unit vector, thanks to Cauchy–Bunyakovsky–Schwarz inequality, we can upper-bound the error of the quadratic form approximation by the error of the $\exp^{-t\mathcal{L}}\mathbf{v}$ approximation, i.e. $|\mathbf{v}^\top f(\mathcal{L})\mathbf{v} - \mathbf{e}_1^\top \mathbf{U} f(\Lambda)\mathbf{U}^\top \mathbf{e}_1| \leq \|\exp^{-t\mathcal{L}}\mathbf{v} - \mathbf{Q}_m \exp^{-t\mathbf{T}_m}\mathbf{e}_1\| = \epsilon_m$.

Following the argumentation in Ubaru et al. (2017), we obtain a condition on the number of Lanczos steps $m$ by setting $\epsilon_m \leq \frac{\epsilon}{2} f_{min}(\lambda)$, where $f_{min}(\lambda)$ is the minimum value of $f$ on $[\lambda_{min}, \lambda_{max}]$. We now derive the absolute error between the Hutchinson estimate of Equation (3) and the SLQ of Equation (6):

$$\left| \operatorname{Tr}_{n_v}(f(\mathcal{L})) - \Gamma \right| = \frac{n}{n_v} \left| \sum_{i=1}^{n_v} \mathbf{v}_i^\top f(\mathcal{L})\mathbf{v}_i - \sum_{i=1}^{n_v} \mathbf{e}_1^\top f(\mathbf{T}^{(i)})\mathbf{e}_1 \right|$$

$$\leq \frac{n}{n_v} \sum_{i=1}^{n_v} \left| \mathbf{v}_i^\top f(\mathcal{L})\mathbf{v}_i - \mathbf{e}_1^\top f(\mathbf{T}^{(i)})\mathbf{e}_1 \right|$$

$$\leq \frac{n}{n_v} \sum_{i=1}^{n_v} \epsilon_m = n\epsilon_m,$$

where $\mathbf{T}^{(i)}$ is the tridiagonal matrix obtained with Lanczos algorithm with starting vector $\mathbf{v}_i$.

$$\left| \operatorname{Tr}_{n_v} f(\mathcal{L}) - \Gamma \right| \leq n\epsilon_m \leq \frac{n\epsilon}{2} f_{min}(\lambda) \leq \frac{\epsilon}{2} \operatorname{Tr}(f(\mathcal{L})), \quad (10)$$

Finally, we formulate SLQ as an $(\epsilon, \delta)$ estimator,

$$1 - \delta \leq \operatorname{Pr}\left[ \left| \operatorname{Tr}(f(\mathcal{L})) - \operatorname{Tr}_{n_v}(f(\mathcal{L})) \right| \leq \frac{\epsilon}{2} \left| \operatorname{Tr}(f(\mathcal{L})) \right| \right]$$

$$\leq \operatorname{Pr}\left[ \left| \operatorname{Tr}(f(\mathcal{L})) - \operatorname{Tr}_{n_v}(f(\mathcal{L})) \right| + \left| \operatorname{Tr}_{n_v}(f(\mathcal{L})) - \Gamma \right| \leq \frac{\epsilon}{2} \left| \operatorname{Tr}(f(\mathcal{L})) \right| + \frac{\epsilon}{2} \left| \operatorname{Tr}(f(\mathcal{L})) \right| \right]$$

$$\leq \operatorname{Pr}\left[ \left| \operatorname{Tr}(f(\mathcal{L})) - \Gamma \right| \leq \epsilon \left| \operatorname{Tr}(f(\mathcal{L})) \right| \right],$$

For the normalized Laplacian $\mathcal{L}$, the minimum eigenvalue is 0 and $f_{\min}(0) = \exp(0) = 1$, hence $\epsilon_m \leq \frac{\epsilon}{2}$, and the eigenvalue interval has $\rho = 0.5$. We can thus derive the appropriate number of Lanczos steps $m$ to achieve error $\epsilon$,

$$
\epsilon \leq \begin{cases} 20e^{-m^2/(2.5t)}, & \sqrt{2t} \leq m \leq t & \text{(11a)} \\ 40t^{-1}e^{-0.5t}\left(\dfrac{0.5et}{m}\right)^m, & m \geq t & \text{(11b)} \end{cases}
$$

Figure 8 shows the tightness of the bound for the approximation of the matrix exponential action on vector $\mathbf{v}$, $\epsilon_m = \|\exp(-t\mathcal{L}) - \mathbf{Q}_m \exp(-t\mathbf{T}_m)\mathbf{e}_1\|$. We can see that for most of the temperatures $t$, very few Lanczos steps $m$ are sufficient, i.e. we can set $m = 10$. However, the error from the Hutchinson estimator dominates the overall error. Figure 9 shows the error of trace estimation does not change with $m$ and for $t = 0.1$ is around $10^{-3}$. In case of a Rademacher $p(\mathbf{v})$, the bound on the number of random samples is $n_v \geq \frac{6}{\epsilon^2} \log(2/\delta)$ (Roosta-Khorasani & Ascher, 2015). Employing 10k vectors results in the error bound of roughly $10^{-2}$. In practice, we observe the performance much better than given by the bound, see Figure 9.

One particular benefit of small $m$ value is that we do not have to worry about the orthogonality loss in the Lanczos algorithm which often undermines its convergence. Since we do only a few Lanczos iterations, the rounding errors hardly accumulate causing little burden in terms of orthogonality loss between the basis vectors of the Krylov subspace.

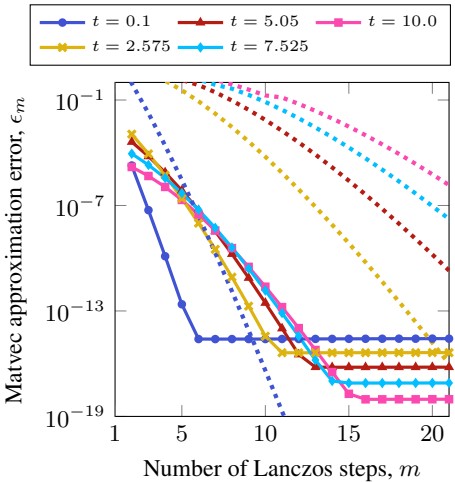

Figure 8: Errors (solid) and error bounds (dotted) for the approximation of matrix exponential action with varying temperature $t$.

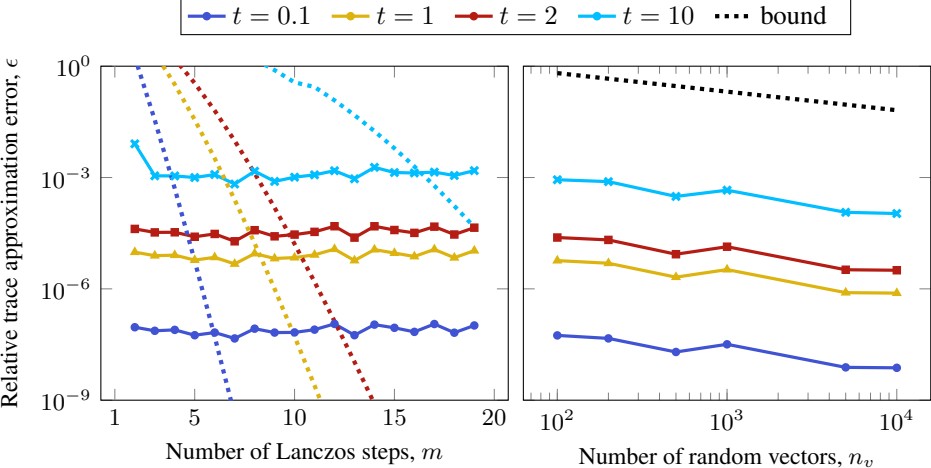

Figure 9: Trace estimation errors (solid) and error bounds (dotted) for: *(left)* the number of Lanczos steps $m$ with fixed number of random vectors $n_v = 100$; *(right)* the number of random vectors $n_v$ in Hutchinson estimator with fixed number of Lanczos steps $m = 10$. Lines correspond to varying temperatures $t$.

## B    VARIANCE REDUCTION

We reduce variance of the randomized estimator through control variates. The idea is to use Taylor expansion to substitute a part of the trace estimate with its easily computed precise value,

$$\text{Tr}(\exp(-t\mathcal{L})) = \texttt{slq}\Big[\exp(-t\mathcal{L}) - (\mathbf{I} - t\mathcal{L} + \frac{t^2\mathcal{L}^2}{2})\Big] + \text{Tr}(\mathbf{I} - t\mathcal{L} + \frac{t^2\mathcal{L}^2}{2}) \tag{12}$$

$$= \texttt{slq}\Big[\exp(-t\mathcal{L}) - (\mathbf{I} - t\mathcal{L} + \frac{t^2\mathcal{L}^2}{2})\Big] + n + \text{Tr}(-t\mathcal{L}) + \frac{t^2\|\mathcal{L}\|_F^2}{2} \tag{13}$$

$$= \texttt{slq}\Big[\exp(-t\mathcal{L})\Big] + \texttt{slq}\Big[t\mathcal{L}\Big] - \texttt{slq}\Big[\frac{t^2\mathcal{L}^2}{2}\Big] - tn + \frac{t^2\|\mathcal{L}\|_F^2}{2}, \tag{14}$$

where we use the fact that $\|\mathcal{L}\|_F = \sqrt{\text{Tr}(\mathcal{L}^\top\mathcal{L})}$ and that the trace of normalized Laplacian is equal to $n$. It does reduce the variance of the trace estimate for smaller temperatures $t \leq 1$.

To obtain this advantage over the whole range of $t$, we utilize the following variance reduction form:

$$\text{Tr}(\exp(-t\mathcal{L})) = \texttt{slq}\Big[\exp(-t\mathcal{L}) - (\mathbf{I} - \alpha t\mathcal{L})\Big] + n(1 - \alpha t), \tag{15}$$

where there exists an alpha that is optimal for every $t$, namely setting $\alpha = 1/\exp(t)$. We can see the variance reduction that comes from this procedure in the Figure 12.

## C    EXPERIMENTS DISCUSSION

Here we include additional results that did not find their way to the main paper body.

### C.1    FID AND KID FAIL TO FIND STRUCTURE IN UNALIGNED CORPORA

Figure 11 shows the matrix of distances for FID and KID aligned and colored in the same way as Figure 2 (a). FID and KID can not find meaningful structure in the data in the same way as IMD as they rely on extrinsic data properties.

### C.2    WORD EMBEDDING EXPERIMENT DETAILS.

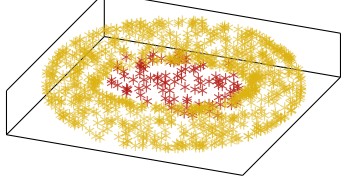

We use gensim (Řehůřek & Sojka, 2010) to learn word vectors on the latest Wikipedia corpus snapshot on 16 languages: Polish, Russian, Greek, Hungarian, Turkish, Arabic, Hebrew, English, Simple English, Swedish, German, Spanish, Dutch, Portugese, Vietnamese, and Waray-Waray. We then compute FID, KID and IMD scores on all the pairs, we average 100 runs for the heatmap figures 2. For the different dimensionality experiment, we learn vectors on the English Wikipedia of sizes equal to the powers of 2 from 4 to 512. After that we compute IMD and covariance error, i.e. normalized PIP loss, between the pairs of sizes to generate the heatmap figure 3.

| metric | good GAN | bad GAN |
|--------|----------|---------|
| FID | 0.00529 ± 0.00070 | 0.00627 ± 0.00076 |
| KID | 0.00172 ± 0.00073 | 0.00259 ± 0.00077 |
| IMD | 9.02059 ± 1.5195 | **14.0732 ±** 2.1706 |

### C.3    VANILLA GAN ON TORUS

We provide an additional experiment clearly showing the case where IMD is superior to its main competitors, FID and KID. We train two vanilla GANs on the points of a 3D torus. The bad

Figure 10:    Bad GAN produces samples inside the torus hole (red). FID and KID cannot detect such behaviour.

GAN fails to learn the topology of the dataset it tries to mimic, yet previous metrics cannot detect this fact. IMD, on the contrary, can tell the difference. Figure 10 shows the points sampled from the GAN with some of the points inside the hole. KID and FID confidence intervals overlap for good and bad GANs, meanwhile IMD scores are clearly distinct from each other.

### C.4    NORMALIZATION DETAILS

For the purpose of normalizing IMD, we need to approximate that graph's eigenvalues. Coja-Oghlan (2007) proved that $\lambda_1 \leq 1 - c\bar{d}^{-1/2} \leq \lambda_2 \leq \lambda_n \leq 1 + c\bar{d}^{-1/2}$ for the core of the graph for some

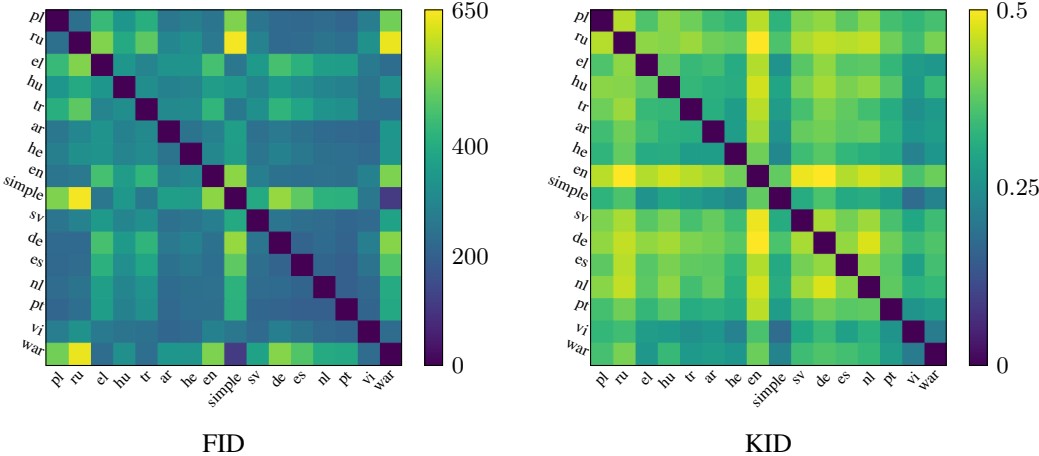

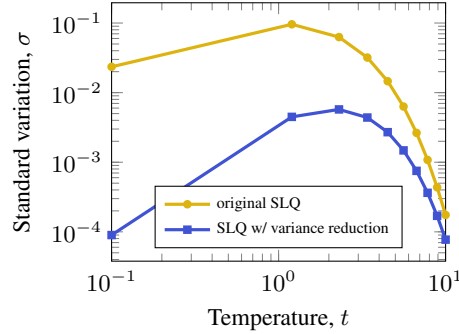

Figure 11: FID and KID are not able to capture language affinity from unaligned word2vec embeddings.

constant $c$. We have empirically found that $c = 2$ provides a tight approximation for random graphs. That coincides with the analysis of Chung et al. (2004), who proved that $\lambda_n = (1 + o(1))2\bar{d}^{-1/2}$ if $d_{\min} \gg \sqrt{\bar{d}} \log^3 n$ even though in our case $d_{\min} = \bar{d} = k$. We thus estimate the spectrum of a random Erdős-Rényi graph as growing linearly between $\lambda_1 = 1 - 2\bar{d}^{-1/2}$ and $\lambda_n = 1 + 2\bar{d}^{-1/2}$, which corresponds to the underlying manifold being two-dimensional (Tsitsulin et al., 2018).

## D EXPERIMENTAL SETTINGS

We train all our models on a single server with NVIDIA V100 GPU with 16Gb memory and $2 \times 20$ core Intel E5-2698 v4 CPU. For the experiment summarized in Table 1 in the Section 4.1 we train WGAN and WGAN-GP models on 4 datasets: MNIST, FashionMNIST, CIFAR10 and CelebA and sample 10k samples, $\mathbf{Y}$, from each of the GANs. We uniformly subsample 10k images from the original datasets, $\mathbf{X}$, and compute the IMD, KID and FID scores between $\mathbf{X}$ and $\mathbf{Y}$. We report the mean as well as the 99% confidence interval across 100 runs.

Figure 12: Variance of the trace estimate.

Below we report the architectures, hyperparameters and generated samples of the models used for the experiments. We train each of the GANs for 200 epochs on MNIST, FMNIST and CIFAR-10, and for 50 epochs on CelebA dataset. For WGAN we use RMSprop optimizer with learning rate of $5 \times 10^{-5}$. For WGAN-GP we use Adam optimizer with learning rate of $10^{-4}$, $\beta_1 = 0.9$, $\beta_2 = 0.999$.

## E GRAPH EXAMPLE

Figure 13 provides visual proof that the 5NN graph reflects the underlying manifold structure of the CIFAR-10 dataset. Clusters in the graph exactly correspond to CIFAR-10 classes.

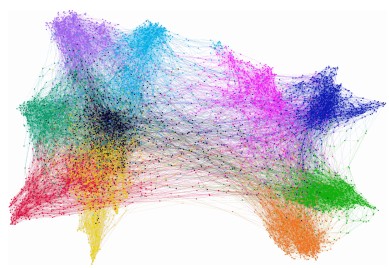

Figure 13: CIFAR-10 graph colored with true class labels.

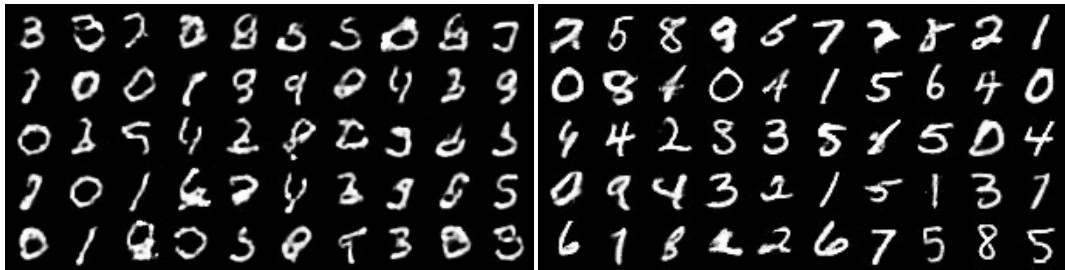

Figure 14: MNIST samples (left: WGAN, right: WGAN-GP)

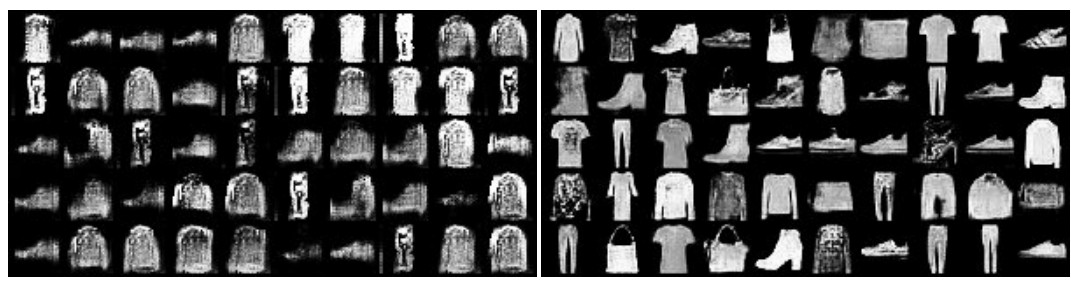

Figure 15: FashionMNIST samples (left: WGAN, right: WGAN-GP)

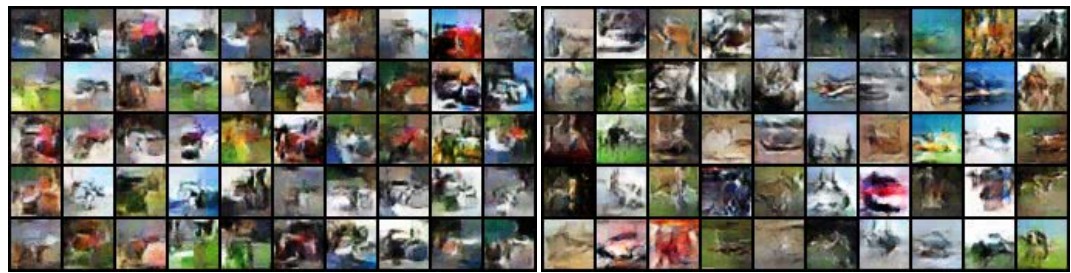

Figure 16: CIFAR-10 samples (left: WGAN, right: WGAN-GP)

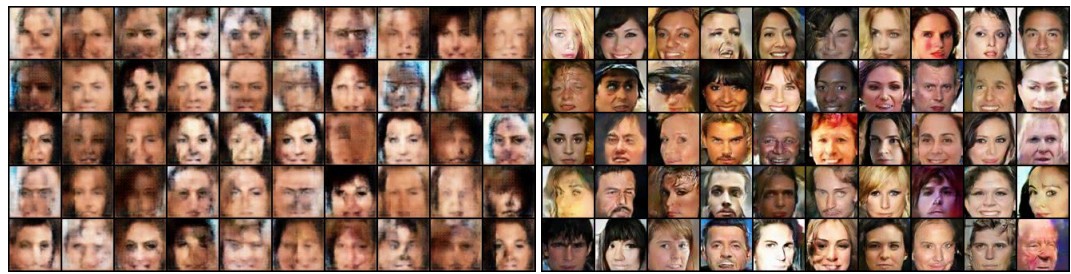

Figure 17: CelebA samples (left: WGAN, right: WGAN-GP)

**MNIST WGAN**

```
ConvGenerator(
  (latent_to_features): Sequential(
    (0): Linear(in_features=100, out_features=512, bias=True)
    (1): ReLU()
  )
  (features_to_image): Sequential(
    (0): ConvTranspose2d(128, 64, kernel_size=(4, 4),
          stride=(2, 2), padding=(1, 1))
    (1): ReLU()
    (2): BatchNorm2d(64, eps=1e-05, momentum=0.1, affine=True)
    (3): ConvTranspose2d(64, 32, kernel_size=(4, 4),
          stride=(2, 2), padding=(1, 1))
    (4): ReLU()
    (5): BatchNorm2d(32, eps=1e-05, momentum=0.1, affine=True)
    (6): ConvTranspose2d(32, 16, kernel_size=(4, 4),
          stride=(2, 2), padding=(1, 1))
    (7): ReLU()
    (8): BatchNorm2d(16, eps=1e-05, momentum=0.1, affine=True)
    (9): ConvTranspose2d(16, 1, kernel_size=(4, 4),
          stride=(2, 2), padding=(1, 1))
    (10): Sigmoid()
  )
)

ConvDiscriminator(
  (image_to_features): Sequential(
    (0): Conv2d(1, 16, kernel_size=(4, 4), stride=(2, 2), padding=(1, 1))
    (1): LeakyReLU(negative_slope=0.2)
    (2): Conv2d(16, 32, kernel_size=(4, 4), stride=(2, 2), padding=(1, 1))
    (3): LeakyReLU(negative_slope=0.2)
    (4): Conv2d(32, 64, kernel_size=(4, 4), stride=(2, 2), padding=(1, 1))
    (5): LeakyReLU(negative_slope=0.2)
    (6): Conv2d(64, 128, kernel_size=(4, 4), stride=(2, 2), padding=(1, 1))
    (7): Sigmoid()
  )
  (features_to_prob): Sequential(
    (0): Linear(in_features=512, out_features=1, bias=True)
    (1): Sigmoid()
  )
)
```

**MNIST WGAN-GP, FMNIST (WGAN, WGAN-GP)**

```
MNISTGenerator(
  (block1): Sequential(
    (0): ConvTranspose2d(256, 128, kernel_size=(5, 5), stride=(1, 1))
    (1): ReLU(inplace)
  )
  (block2): Sequential(
    (0): ConvTranspose2d(128, 64, kernel_size=(5, 5), stride=(1, 1))
    (1): ReLU(inplace)
  )
  (deconv_out): ConvTranspose2d(64, 1, kernel_size=(8, 8), stride=(2, 2))
  (preprocess): Sequential(
    (0): Linear(in_features=128, out_features=4096, bias=True)
    (1): ReLU(inplace)
  )
  (sigmoid): Sigmoid()
)

MNISTDiscriminator(
  (main): Sequential(
    (0): Conv2d(1, 64, kernel_size=(5, 5), stride=(2, 2), padding=(2, 2))
    (1): ReLU(inplace)
    (2): Conv2d(64, 128, kernel_size=(5, 5), stride=(2, 2), padding=(2, 2))
    (3): ReLU(inplace)
    (4): Conv2d(128, 256, kernel_size=(5, 5), stride=(2, 2), padding=(2, 2))
    (5): ReLU(inplace)
  )
  (output): Linear(in_features=4096, out_features=1, bias=True)
)
```

**CIFAR-10 (WGAN, WGAN-GP)**

```
CIFARGenerator(
  (preprocess): Sequential(
    (0): Linear(in_features=128, out_features=4096, bias=True)
    (1): BatchNorm1d(4096, eps=1e-05, momentum=0.1, affine=True)
    (2): ReLU(inplace)
  )
  (block1): Sequential(
    (0): ConvTranspose2d(256, 128, kernel_size=(2, 2), stride=(2, 2))
    (1): BatchNorm2d(128, eps=1e-05, momentum=0.1, affine=True)
    (2): ReLU(inplace)
  )
  (block2): Sequential(
    (0): ConvTranspose2d(128, 64, kernel_size=(2, 2), stride=(2, 2))
    (1): BatchNorm2d(64, eps=1e-05, momentum=0.1, affine=True)
    (2): ReLU(inplace)
  )
  (deconv_out): ConvTranspose2d(64, 3, kernel_size=(2, 2), stride=(2, 2))
  (tanh): Tanh()
)

CIFARDiscriminator(
  (main): Sequential(
    (0): Conv2d(3, 64, kernel_size=(3, 3), stride=(2, 2), padding=(1, 1))
    (1): LeakyReLU(negative_slope=0.01)
    (2): Conv2d(64, 128, kernel_size=(3, 3), stride=(2, 2), padding=(1, 1))
    (3): LeakyReLU(negative_slope=0.01)
    (4): Conv2d(128, 256, kernel_size=(3, 3), stride=(2, 2), padding=(1, 1))
    (5): LeakyReLU(negative_slope=0.01)
  )
  (linear): Linear(in_features=4096, out_features=1, bias=True)
)
```

### CelebA (WGAN, WGAN-GP)

```
CelebaGenerator(
  (preprocess): Sequential(
    (0): Linear(in_features=128, out_features=8192, bias=True)
    (1): BatchNorm1d(8192, eps=1e-05, momentum=0.1, affine=True)
    (2): ReLU(inplace)
  )
  (block1): Sequential(
    (0): ConvTranspose2d(512, 256, kernel_size=(5, 5), stride=(2, 2),
            padding=(2, 2), output_padding=(1, 1), bias=False)
    (1): BatchNorm2d(256, eps=1e-05, momentum=0.1, affine=True)
    (2): ReLU(inplace)
  )
  (block2): Sequential(
    (0): ConvTranspose2d(256, 128, kernel_size=(5, 5), stride=(2, 2),
            padding=(2, 2), output_padding=(1, 1), bias=False)
    (1): BatchNorm2d(128, eps=1e-05, momentum=0.1, affine=True)
    (2): ReLU(inplace)
  )
  (block3): Sequential(
    (0): ConvTranspose2d(128, 64, kernel_size=(5, 5), stride=(2, 2),
            padding=(2, 2), output_padding=(1, 1), bias=False)
    (1): BatchNorm2d(64, eps=1e-05, momentum=0.1, affine=True)
    (2): ReLU(inplace)
  )
  (deconv_out): ConvTranspose2d(64, 3, kernel_size=(5, 5), stride=(2, 2),
          padding=(2, 2), output_padding=(1, 1))
  (tanh): Tanh()
)

CelebaDiscriminator(
  (main): Sequential(
    (0): Conv2d(3, 64, kernel_size=(5, 5), stride=(2, 2), padding=(2, 2))
    (1): LeakyReLU(negative_slope=0.01)
    (2): Conv2d(64, 128, kernel_size=(5, 5), stride=(2, 2), padding=(2, 2))
    (3): LeakyReLU(negative_slope=0.01)
    (4): Conv2d(128, 256, kernel_size=(5, 5), stride=(2, 2), padding=(2, 2))
    (5): LeakyReLU(negative_slope=0.01)
    (6): Conv2d(256, 512, kernel_size=(5, 5), stride=(2, 2), padding=(2, 2))
    (7): LeakyReLU(negative_slope=0.01)
    (8): Conv2d(512, 1, kernel_size=(4, 4), stride=(1, 1))
  )
)
```

