# OpenReview forum: "The Shape of Data: Intrinsic Distance for Data Distributions"
_ICLR.cc/2020/Conference — Accept (Poster)_

### Official Review · AnonReviewer3 · 2019-10-13
**Official Blind Review #3**

**Rating:** 6

**Review:**

IMD is a lower bound to Gromov-Wasserstein distance. Which implies that when IMD is small, this does not guarantee that GW will be small. It is interesting how large can be that gap (and when typically the gap increases).
The term e{−2(t+t^−1)} in the definitions of GW and IMD, obviously imposes some "scale" for both manifolds M and N, so it is hard to call IMD multi-scale.

Experiments with language affinities are not convincing to me. How is armenian and hebrew are close, or hungarian and turkish (if "el" is greek, then greek, hungarian and turkish are close somehow?).


**Experience Assessment:**

I have read many papers in this area.

**Review Assessment: Checking Correctness Of Derivations And Theory:**

I did not assess the derivations or theory.

**Review Assessment: Checking Correctness Of Experiments:**

I assessed the sensibility of the experiments.

**Review Assessment: Thoroughness In Paper Reading:**

I made a quick assessment of this paper.

---

> ### Author Response · Authors · 2019-11-14
> **Response to Official Blind Review #3**
>
> We thank the reviewer for the detailed comments. Please find our responses below.
>
> With respect to bounding the Gromov-Wasserstein distance, the bound is the loosest for isospectral manifolds / graphs. While there are known isospectral constructions for both Riemmanian manifolds [1,2] and graphs [3], it is conjectured that the proportion of isospectral graphs drops to zero with increasing number of nodes [4].
>
> Regarding the restrictions of scale due to the normalization term $e^{-2(t+t^{-1})}$, this term prevents the blowup of heat kernel trace as the scale parameter $t$ approaches 0 from the right. In practice, this temperature scale ($t \approx 0$) does not bring any information except the difference in the number of nodes in the graph (samples in the data), i.e. $\sum_i^n e^{-0\lambda_i} = n$. Going further away from 0, we can observe this behaviour from the Taylor expansion of the matrix exponential of the heat kernel trace for kNN graphs $h_t \approx n - tn + t^2n$. We empirically observe that when $t<0.5$ first two terms of the Taylor expansion approximate the heat trace with a very small error ($<10^{-3}$) with no information that is "interesting" in terms of the shape.
>
> References:
>
> [1] J. Milnor," Eigenvalues of the Laplace operator on certain manifolds", Proc. Natl. Acad. Sci. USA 51 (1964)
>
> [2] T. Sunada, "Riemannian coverings and isospectral manifolds", Annals of Mathematics (1985)
>
> [3] S. Butler, and J. Grout. "A construction of cospectral graphs for the normalized Laplacian." The Electronic Journal of Combinatorics 18.1 (2011)
>
> [4] R. C. Wilson, and P. Zhu. "A study of graph spectra for comparing graphs and trees." Pattern Recognition 41.9 (2008)
>
> Regarding your question about language similarities depicted in the Figure 2, in the submitted version language names were only discussed in the Appendix; we have moved the language list to the caption in the updated version of the paper. We apologize for the confusion in the submitted draft.
>
> Arabic (not Armenian) is a Semitic language close to Hebrew. Thanks to your question, we can highlight that we have detected not only language families/subfamilies (such as Slavic, Romance, and Semitic), but also linguistic convergence areas (or Sprachbunds) [1]. Hungarian and Turkish have been in contact in the Eurasian Ural-Altaic convergence area [2], and more recently in the Ottoman period [3]. Greek and Turkish are close as members of the Balkan Sprachbund [4]. Hungarian has been in contact with Greek during the establishment of the Kingdom of Hungary (cf. the Greek inscriptions in the Holy Crown of Hungary, National Museum, Budapest, which incidentally refers to Hungary as "Tourkia", suggesting an Ural-Altaic connection as noted [5]).
>
> References:
>
> [1] https://en.wikipedia.org/wiki/Language_convergence
>
> [2] https://en.wikipedia.org/wiki/Ural\%E2\%80\%93Altaic_languages
>
> [3] https://en.wikipedia.org/wiki/Ottoman_Hungary
>
> [4] https://en.wikipedia.org/wiki/Balkan_sprachbund
>
> [5] https://en.wikipedia.org/wiki/Holy_Crown_of_Hungary

---

### Official Review · AnonReviewer1 · 2019-10-18
**Official Blind Review #1**

**Rating:** 6

**Review:**

The paper propose a novel way to measure similarity between datasets, which e.g. is useful to determine if samples from a generative model resembles a test dataset. The approach is presented as an efficient numerical algorithm for working with diffusion on data manifolds.

I am very much in doubt about this paper as there are too many aspects of the work I do not (currently) understand. I suspect that the underlying idea is solid, but in its current form I cannot recommend publication of the paper.

Detailed questions and comments:

*) It seems that the focus is on GANs and related models that do not come with a likelihood. In my reading, the practical problem that the paper address is model comparison in models that do not have a likelihood. Is that correct? If so, I am left to wonder why models with a likelihood a not discussed, and why the such models aren't included in the experiments? Wouldn't the likelihood be a sensible baseline?

*) The term "multi-scale" is not defined until page 3. I'd recommend defining this term much more early or avoid its use until its defined.

*) I found the geometric exposition to be rather confusing. In Sec. 3.1 emphasis seem to initially be on Euclidean diffusion, and at some point emphasis changes to diffusion on graphs. Sometimes the paper seem focused on diffusion on general manifolds. I found this exposition to be rather chaotic, and I suspect that this is the root cause of me not understanding many aspects of the work.

*) In Sec. 3.2 the notion of diffusion *on* manifolds is discussed. At some point (not quite clear to me when) the results of this discussion is applied to diffusion *between* manifolds. I don't quite understand what it means to diffuse between manifolds. I would have appreciated a more explicit exposition here.

*) In the concluding remarks it is stated that IMD provides guarantees. Which guarantees are that?


**Experience Assessment:**

I have published one or two papers in this area.

**Review Assessment: Checking Correctness Of Derivations And Theory:**

I assessed the sensibility of the derivations and theory.

**Review Assessment: Checking Correctness Of Experiments:**

I assessed the sensibility of the experiments.

**Review Assessment: Thoroughness In Paper Reading:**

I made a quick assessment of this paper.

---

> ### Author Response · Authors · 2019-11-14
> **Response to Official Blind Review #1**
>
> We thank the reviewer for the detailed questions and comments. We hope that our response and improved exposition of the paper will address the concerns.
>
> [#1] We consider a general problem of comparing the geometry of samples drawn from unknown data distributions, including cases without any model and, consequently, without likelihood. In our experiments, we study a diverse set of applications, focusing on the scenarios that were not previously possible in the literature. For example, in the experiment in Section 4.2 we compare word embeddings across languages with no model attached to it and unaligned vocabularies. In a different vein, in Section 4.3 we consider similarities in representations for networks *across datasets*, this scenario is not possible to study using traditional methods such as [2, 3]. Besides, even for likelihood-based models, the objective has been shown to not always be sensible for out-of-distribution examples [1].
>
> [#2,#3] We have improved the text for a better exposition. The discussion in Section 3.1 starts with heat diffusion in the Euclidean space to present the most common space in the machine learning literature. We shift the focus to graph Laplacians constructed from the samples from general manifolds since they are known to approximate their diffusion properties (see Theorem 1). We then approximate graphs’ spectrum with stochastic Lanczos quadrature method and use the approximated spectral descriptors to compute the lower bound to the Gromov-Wasserstein distance between the underlying manifolds.
>
> [#4] The diffusion process does **not** occur between manifolds. Instead, to compare manifolds we compute the distance **between the descriptors** of these manifolds obtained by the diffusion inside each of them. Such a distance lower bounds an optimal matching of the heat kernel defined on each manifold and thus the Gromov-Wasserstein distance between manifolds.
>
> [#5] IMD inherits several guarantees from past literature, and, in addition, we provide approximation error guarantees in Theorem 2.  IMD is supported by multiple theoretical statements, while not all of them are error guarantees, we make sure all the steps in the algorithm are principled. IMD is an approximation of the lower bound of the spectral Gromov-Wasserstein distance, as per bounds of (Mémoli, 2011). We use an OR-graph construction to approximate the Laplace-Beltrami operator of the underlying data manifold, as per (Ting et al., 2010). Then, we approximate the lower bound with Stochastic Lanczos Quadrature, and do so with error controlled by our Theorem 2.
>
> [1] E. Nalisnick, et al. "Do deep generative models know what they don't know?." ICLR (2019)
>
> [2] A. Morcos, M. Raghu, and S. Bengio. "Insights on representational similarity in neural networks with canonical correlation." NeurIPS (2018)
>
> [3] S. Kornblith, et al. "Similarity of neural network representations revisited." ICML (2019)

---

> > ### Comment · AnonReviewer1 · 2019-11-15
> > **Rebuttal acknowledged**
> >
> > Thanks for the detailed feedback, which cleared up a few things for me (in particular #4 was a big deal for me). I will change my score to a weak accept.

---

### Official Review · AnonReviewer2 · 2019-10-25
**Official Blind Review #2**

**Rating:** 6

**Review:**

1. Define/explain data manifold: in the abstract, ``data moments" suggests the data is treated as random variables; the authors could explain more on how they make the connection between random variables and manifolds
2. introduction: include an example of ``models having different representation space"
3. introduction: elaborate more on ``unaligned data manifolds"
4. introduction: the distinction between ``intrinsic" and ``extrinsic" can be made more explicit; as currently written, the difference between ``extrinsic"/``multi-scale"/``utilizing higher moments" isn't clear
5. the authors mentioned the hypothesis that high-dimensional data lies on a low-dimensional manifold, but in (3.1) they only considered $\mathbb{R}^d$ without justifying this restriction
6. (3.1) ``$u(t, \cdot) \to \delta(\cdot -x)$" change the dot to a variable
7. (3.1) ``$u(t, x) = 0 \forall x \in \partial X$ and for all $t$" change to ``for all $t$ and $x \in \partial X$
8. equation 1, write ``we restrict our exposition to Euclidean spaces $X = \mathbb{R}^d$" and remove ``$\forall x, x' \in \mathbb{R}^d, t \in \mathbb{R}^+$"
9.  (3.1) the phrase ``a compact X including $\mathbb{R}^d$" is confusing
    as $\mathbb{R}^d$ is not compact under Euclidean topology
10.  (3.1) ``for a local domain with Dirichlet condition $D$" change to ``for a local domain $D$ with Dirichlet condition"
11.  (3.1) ``Definition 1" is more of a proposition/theorem.
12.  (3.1) notation: in the body paragraph the authors use $k(x, x', t)$ but in ``definition 1" they used $k(x, y, t)$
13. (3.1) ``$c$ disconnected components" to ``$c$ connected components"
14. (3.1) the authors could elaborate more about how HKT contains all the information in the graph's spectrum. Currently, they only stated the limit equal to the number of connected components.
15. Github code is helpful but it would be better if they can include an algorithm section
16. A figure of the architecture could be included under the ``putting IMD together" section

**Experience Assessment:**

I have published one or two papers in this area.

**Review Assessment: Checking Correctness Of Derivations And Theory:**

I assessed the sensibility of the derivations and theory.

**Review Assessment: Checking Correctness Of Experiments:**

I assessed the sensibility of the experiments.

**Review Assessment: Thoroughness In Paper Reading:**

I read the paper at least twice and used my best judgement in assessing the paper.

---

> ### Author Response · Authors · 2019-11-14
> **Response to Official Blind Review #2**
>
> We thank the reviewer for the positive feedback and constructive suggestions. We have incorporated points #4-#13  into the updated version of the paper, and will continue to work on better presentation of the work after this comment is posted, i.e. we will include an algorithm and update the introduction as requested. We discuss some of the points below for better exposition.
>
> [#1] In machine learning it is common to treat the data as points sampled from some unknown distribution in $\mathbb{R}^d$. We mention data moments to contrast with existing techniques for measuring closeness between two samples. We propose to make use of the geometry of samples instead. We treat samples as coming from some manifold embedded in $\mathbb{R}^d$ (possibly equipped with its own distribution).
>
> [#4]  An **extrinsic property** undergoes a change when transformations like rotation of shifts are applied to data. For example, the mean of a Gaussian is not invariant under shift, but variance is. An **intrinsic property** stays invariant under **all** isometric transformations (dependent only on the metric tensor). The intuition behind **multi-scale property** of the heat kernel is closely connected with time parameter t. As t grows larger, heat kernel depends on larger neighborhoods of points, so the heat kernel values can serve as a description/summary of the neighborhood at different scales. This is similar to the high-order moments of the data if we treat it in a traditional way as having an unknown distribution in $\mathbb{R}^d$ from which it is sampled. The more moments of a distribution we know, the more we know about its shape. In a similar way the heat kernel accumulates the shape information of the manifold covering all the possible moments of the data seen as a distribution.
>
> [#14] We mention that eigenvalues of the graph/shape can be inferred from the spectrum. The exact procedure is defined in the remark 4.8 in (Mémoli, 2011); we have added this reference to the main body of the paper. In short, we iteratively seek eigenvalues $\lambda_i$ such that $\lim_{t \rightarrow \infty}{e^{\lambda_i t} hkt(t)} \not= 0$, starting from $\lambda_0$. This allows to establish an equivalence relation between the spectrum and the heat kernel trace, allowing us to make precise statements about the informativeness of our metric.

---

### Author Response · Authors · 2019-10-06
**Formatting corrections to Figure 4**

Due to an error in figure generation, Figure 4(b) presents incorrect relative accuracy for VGG-16 trained on the CIFAR-100 dataset.

Correct Figure 4 is available at the following URL: https://i.imgur.com/bMi0tl8.png
We apologize for the error.

---

### Decision · Program_Chairs · 2019-12-19

**Decision:**

Accept (Poster)

**Comment:**

This paper introduces a way to measure dataset similarities. Reviewers all agree that this method is novel and interesting. A few questions initially raised by reviewers regarding models with and without likelihood, geometric exposition, and guarantees around GW, are promptly answered by authors, which raised the score to all weak accept.